# The impact of hospital pharmacists' psychological contracts on medication adherence management: The mediating role of job burnout

Yongyong Luo[1,2,3◉], Mei Nie[2,4◉], Cheng Chen[1,3,4], Yong He[4,5], Xiaoyu Jiang[4,5], Jianhua Tang[6], Ting Zhang[4,7], Yang Gu[4,8], Fushan Tang[1,3,4]*

1 Department of Clinical Pharmacy, Key Laboratory of Basic Pharmacology of Guizhou Province and School of Pharmacy, Zunyi Medical University, Zunyi, Guizhou Province, China, 2 Department of Pharmacy, Bijie Maternal and Child Health Hospital, Bijie, Guizhou Province, China, 3 Key Laboratory of Basic Pharmacology of Ministry of Education and Joint International Research Laboratory of Ethnomedicine of Ministry of Education, Zunyi Medical University, Zunyi, Guizhou Province, China, 4 The Key Laboratory of Clinical Pharmacy in Zunyi City, Zunyi Medical University, Zunyi, Guizhou Province, China, 5 Department of Pharmacy, The Second Affiliated Hospital of Zunyi Medical University, Zunyi, Guizhou Province, China, 6 Cancer Research UK Manchester Institute, The University of Manchester, Cheshire, Manchester, United Kingdom, 7 Department of Pharmacy, Guiyang Hospital of Stomatology, Guiyang, Guizhou Province, China, 8 Department of Pharmacy, Hezhang County People's Hospital, Bijie, Guizhou Province, China

◉ These authors contributed equally to this work.
* fstang@vip.163.com

## Abstract

### Background

This study examines the relationship between hospital pharmacists' psychological contracts and their attitudes and behaviors in managing patient medication adherence. The goal is to provide a solid scientific foundation and practical recommendations for improving pharmacists' involvement in adherence management.

### Methods

A Delphi evaluation, involving pharmacy experts from healthcare institutions, was conducted to refine the Pharmacists' Attitudes and Behaviors Toward Patient Medication Adherence Management Scale. This scale, alongside other relevant instruments, was used to survey pharmacists working in public healthcare institutions across two cities in Guizhou, China.

### Results

The Attitudes Toward Patient Medication Adherence Management Scale exhibited strong reliability, with a Cronbach's α coefficient of 0.796 and a KMO value of 0.899. The Behaviors Toward Patient Medication Adherence Management Scale demonstrated even higher reliability, with a Cronbach's α of 0.986 and a KMO value of

**Data availability statement:** Relevant data supporting the findings of this study can be made available upon reasonable request. In addition to contacting the corresponding author Professor Fushan Tang (Email: fstang@zmu.edu.cn), data access requests may also be directed to Deputy Chief Pharmacist Chengchen Yin at Renhuai Traditional Chinese Medicine Hospital (Email: 824196010@qq.com), who serves as an institutional representative not listed as an author. To ensure persistent and long-term availability, all study data are securely stored in two independent locations: the Clinical Pharmacy Office of Zunyi Medical University and Renhuai Traditional Chinese Medicine Hospital.

**Funding:** This study was supported by the Special Funds for Science and Technology Cooperation in Guizhou Province and Zunyi City (Shengshikehe 2015] 53) and the Graduate Student Research Fund of Zunyi Medical University (ZYK233). the funders had no role in study design, data collection and analysis, decision to publish, or preparation of the manuscript. They provided financial support only.

**Competing interests:** The authors have declared that no competing interests exist.

0.963. A significant positive correlation was found between pharmacists' psychological contracts and their attitudes and behaviors toward patient medication adherence management ($P < 0.01$). Additionally, job burnout was identified as a partial mediator, accounting for 23.41% of the total effect; Targeted interventions to strengthen psychological contracts should be implemented to improve pharmacists' motivation and performance, ultimately boosting patient safety and treatment outcomes.

## Conclusions

The fulfillment of pharmacists' psychological contracts plays a key role in enhancing their involvement in medication adherence management. Targeted interventions to strengthen these psychological contracts should be implemented to improve pharmacists' motivation and performance, ultimately boosting patient safety and treatment outcomes.

## Introduction

Medication adherence remains a critical issue in healthcare, with non-adherence persisting as a widespread and enduring challenge [1]. The World Health Organization (WHO) estimates that 25% to 50% of patients worldwide fail to follow prescribed medication regimens. In the United States, inadequate adherence is responsible for around 125,000 deaths annually—equivalent to the combined mortality from colorectal, breast, and prostate cancers [2]. Poor adherence undermines treatment efficacy, leading to incomplete therapy, disease progression, and, in some cases, death. A study on patients with chronic obstructive pulmonary disease (COPD) found that poor adherence worsened symptoms, increased hospitalization and mortality, and raised healthcare costs [3]. As healthcare models evolve and patient needs grow more complex, the role of pharmacists in enhancing medication adherence and ensuring patient safety has become increasingly important [4–6]. Within the current healthcare system in China, patient medication adherence management is generally not considered part of the formal duties of most pharmacists, but rather constitutes an additional, proactive professional service. Specifically, the core responsibilities of hospital pharmacists typically focus on prescription review, dispensing, medication counseling, and drug management. Activities related to adherence management—such as providing medication guidance, conducting follow-up monitoring, and intervening in inappropriate medication use—have not yet been formally incorporated into job descriptions or performance evaluations in many hospitals. While some clinical pharmacists or pharmacy service teams may proactively engage in adherence management, in most cases this relies on voluntary effort, requiring extra time and energy. Although national and local pharmacy service policies are gradually encouraging pharmacists to participate in medication therapy management, including monitoring, guidance, and adherence interventions, in practice adherence management remains largely a "non-mandatory but critical" professional service. Accordingly, we hypothesize that pharmacists' participation in patient medication adherence management

may be driven by their psychological contract perceptions; specifically, the stronger a pharmacist's perception of their psychological contract, the more willing they are to actively engage in adherence management.

The concept of the psychological contract refers to an employee's perception of mutual obligations between themselves and their organization, first introduced by Argyris in the 1960s [7]. This implicit agreement significantly influences employee engagement [8], and research highlights that fulfilling the psychological contract enhances motivation [9]. Previous studies have shown that pharmacists' psychological contracts positively impact their attitudes and behaviors in providing pharmaceutical services [10]. However, unmet organizational commitments can breach the psychological contract, leading to reduced job satisfaction, diminished motivation, and increased turnover intentions. Research on physicians shows that such breaches elicit strong negative emotions, resulting in undesirable behaviors [11,12]. To enhance employee motivation and loyalty, organizations must prioritize psychological contracts alongside economic agreements, thereby improving healthcare quality [13]. Understanding pharmacists' psychological contracts is essential to fostering engagement, improving medication adherence, and ultimately enhancing patient care quality.

Job burnout has been identified as an important mediating factor linking psychological contracts with work outcomes. According to social exchange theory and conservation of resources theory, breaches in psychological contracts can deplete employees' psychological resources, thereby increasing burnout, which in turn negatively influences professional performance and patient care quality [14,15]. At the same time, existing literature emphasizes that healthcare professionals' engagement and well-being are closely tied to both psychological contract fulfillment and their role in ensuring effective patient medication adherence [16]. Thus, investigating the interrelationships among psychological contracts, medication adherence management, and job burnout is theoretically grounded in these frameworks and empirically supported by prior studies.

In our previous research, we developed and validated a psychological contract scale for hospital pharmacists based on data from 77 public medical institutions in Zunyi, China [17]. This scale was proven reliable and valid, providing a robust tool for future studies. The current study aims to explore the relationship between hospital pharmacists' psychological contracts and their attitudes and behaviors toward patient medication adherence. By identifying the psychological contract factors influencing pharmacists' involvement in adherence management, this research aims to offer insights for improving their participation and ultimately enhancing patient adherence and health outcomes.

## Materials and methods

This study was conducted in two phases. In the first phase, a scale measuring hospital pharmacists' attitudes and behaviors toward medication adherence management was developed and psychometrically validated. In the second phase, the validated scale was applied to investigate factors influencing pharmacists' participation in medication adherence management.

### Phase I: Scale development and validation

**2.1.1. Retrieve Literature.** A comprehensive literature search was conducted using the keywords "Medication Adherence," "Pharmacist," "Research Progress," "Psychological Contract," and "Management" in both Chinese and English. Several databases were queried, including CNKI, Wanfang, VIP, PubMed, and Web of Science. The literature review had an exploratory nature, with the primary aim of developing an initial item pool, which was subsequently refined and optimized through expert consultation. Based on expert consultations, a preliminary draft of the "Pharmacist's Attitudes and Behaviors Toward Patient Medication Adherence Management" scale was developed. This scale comprises two subscales: "Pharmacist's Attitudes Toward Patient Medication Adherence Management" (9 items) and "Pharmacist's Behaviors Toward Patient Medication Adherence Management" (24 items).

**2.1.2. Delphi expert consultation.** A Delphi consultation questionnaire was designed based on the initial scale, consisting of three parts: (1) Introduction (study background, purpose, and significance), (2) Main questionnaire (item content details), and (3) Expert demographic survey (expert information, item familiarity, and judgment criteria).

Sixteen experts were selected for the Delphi process, with inclusion criteria: (1) pharmacists in healthcare institutions, (2) associate senior title or higher, (3) bachelor's degree or above, (4) at least 5 years of experience, and (5) informed consent to participate. Exclusion criteria included external trainers.

**2.1.3. Expert evaluation and data analysis.** Expert characteristics were summarized using frequencies and percentages. Expert engagement and authority were assessed through the response rate and expert authority coefficient (Cr), calculated as follows:

$$Cr = \frac{Cs + Ca}{2}$$

A Cr value ≥ 0.7 indicates good reliability. Here, Cs represents the expert's familiarity with the questionnaire (ranging from "very familiar" (0.9) to "very unfamiliar" (0.1)) [18], and Ca refers to the expert's judgment basis, which is classified into practical experience, theoretical analysis, relevant literature, and personal intuition, with corresponding values assigned as shown in Table 1 [19].

The concentration of expert opinions was assessed using the importance score and the percentage of maximum scores. Importance was rated on a 5-point Likert scale (1 = very unimportant to 5 = very important). Consistency among expert opinions was evaluated through the coefficient of variation (CV) and Kendall's coefficient of concordance (W). Specifically: CV < 0.3 indicates high coordination; W with P < 0.05 indicates significant agreement [20].

Items were retained if the average importance score was ≥ 3.5 and the CV was < 0.3 [21,22]. Expert opinions were also used to determine whether each item should be retained in the final version.

**2.1.4. Reliability and validity testing.** Reliability was assessed using Cronbach's α coefficient, where values between 0.5 and 0.7 indicate acceptable reliability, values above 0.7 indicate high reliability, and values above 0.9 indicate excellent reliability [23].

Validity was assessed using exploratory factor analysis (EFA). The Kaiser-Meyer-Olkin (KMO) test and Bartlett's test of sphericity were conducted. A KMO value > 0.7 indicates suitability for factor analysis, while a significant P-value (P < 0.05) from Bartlett's test confirms data appropriateness for analysis [24].

## Phase II: Influencing factors analysis

**2.2.1. Survey instrument.** The present study primarily employed a paper-based questionnaire combined with on-site interviews. When necessary, electronic questionnaires were distributed via email, postal mail, or professional

**Table 1. Expert Judgment Basis Quantification Table.**

| The basis for expert review | Expert Scoring Judgment Coefficient | | |
|---|---|---|---|
| | large | middle | little |
| Practical experience | 0.5 | 0.4 | 0.3 |
| Theoretical analysis | 0.3 | 0.2 | 0.1 |
| Relevant literature | 0.1 | 0.1 | 0.1 |
| Personal intuition | 0.1 | 0.1 | 0..1 |

online survey platforms such as "Wenjuanxing" to supplement data collection. The questionnaire consisted of four parts: participants' demographic information, the Hospital Pharmacists' Psychological Contract Scale, the Maslach Burnout Inventory, and the Pharmacists' Attitudes and Behaviors Toward Patient Medication Adherence Management scale.

The Hospital Pharmacists' Psychological Contract Scale was developed in our previous research [17] and comprises three subscales assessing the responsibilities of the government/society, the hospital, and the pharmacists themselves, with a total of 40 items. The Kaiser–Meyer–Olkin (KMO) values for the three subscales were 0.957, 0.930, and 0.917, respectively, all exceeding the recommended threshold of 0.6, and Bartlett's test of sphericity was statistically significant ($p < 0.001$), indicating the suitability of the data for factor analysis. Both Cronbach's α coefficients and split-half reliability coefficients were above 0.6, demonstrating acceptable reliability and validity of the scale. All items are rated on a 7-point Likert scale, where 0 indicates "not the responsibility of the hospital/pharmacist/government or society," 1 indicates "responsibility exists but not fulfilled," 2 indicates "mostly unfulfilled," 3 indicates "partially fulfilled," 4 indicates "fulfilled halfway," 5 indicates "mostly fulfilled," and 6 indicates "fully fulfilled." The total score ranges from 0 to 240..

The Maslach Burnout Inventory (service sector version) was used, translated and adapted by Li Chaoping [25]. It consists of 22 items rated on a 7-point Likert scale (1 = never, 2 = rarely, 3 = occasionally, 4 = often, 5 = frequently, 6 = very frequently, 7 = daily), with higher scores indicating higher levels of burnout. Items 4, 7, 9, 12, 17–19, and 21 are reverse-scored, meaning that higher scores on these items reflect lower levels of burnout. The internal consistency coefficients (Cronbach's α) for the three dimensions were 0.89, 0.79, and 0.87, respectively, indicating good reliability.

**2.2.2. Participant recruitment.** A stratified random cluster sampling method was employed to recruit participants from tertiary, secondary, and primary healthcare institutions in Zunyi and Bijie, China. The study was conducted between November and December 2024. All participants were fully informed of the study's purpose, voluntarily agreed to participate, and provided verbal informed consent. Inclusion criteria included: (1) a background in pharmacy, (2) current employment as a pharmacist, and (3) voluntary participation. Trainee pharmacists were excluded from the study. No financial or material incentives were provided for participation.

**2.2.3. Statistical analysis.** Data were entered into Excel and analyzed using SPSS 29.0. A P-value < 0.05 was considered statistically significant, with all tests being two-tailed. Demographic data were presented as frequencies and percentages, and scale scores as mean ± standard deviation. For binary and multi-category data, independent sample t-tests, one-way ANOVA, and Kruskal-Wallis H tests were used, with LSD and Games-Howell post hoc tests for multiple comparisons. Spearman's correlation analysis was used to assess relationships between pharmacists' psychological contract, job burnout, and their attitudes and behaviors toward medication adherence management. Multiple linear regression was used to identify factors influencing pharmacists' participation in medication adherence management.

**2.2.4. Ethical approval.** This study follows the Declaration of Helsinki and was approved by the Ethics Committee of Zunyi Medical University (ZMCER [2023] 1–008). All participants were fully informed about the purpose of the study, they obtained verbal informed consent, and their participation was voluntary and anonymous, as described in the pre-questionnaire letter and the survey instructions.

## Results

### Phase I Results: Scale development and validation

#### Expert panel.

#### Expert demographics.
All 16 experts in the Delphi consultation held at least a bachelor's degree, with 25% holding doctoral degrees. All experts held professional titles of associate senior or higher, and 44% had over 20 years of work experience (Table 2).

**Table 2. Demographic Characteristics of Delphi Experts.**

| Group | Basic Information | Number | Percentage |
|---|---|---|---|
| Gender | Male | 7 | 44% |
| | Female | 9 | 56% |
| Age | 30-39 | 5 | 31% |
| | 40-49 | 7 | 44% |
| | 50-59 | 3 | 19% |
| | 60 and above | 1 | 6% |
| Years of Experience | Less than 10 years | 0 | 0% |
| | 10-20 years | 9 | 56% |
| | More than 20 years | 7 | 44% |
| Education | Doctorate | 4 | 25% |
| | Master's | 8 | 50% |
| | Bachelor's | 4 | 25% |
| Professional Title | Senior Title | 9 | 56% |
| | Associate Senior Title | 7 | 44% |
| Research Field | Pharmacy | 1 | 6% |
| | Clinical Pharmacy | 10 | 64% |
| | Hospital Pharmacy | 1 | 6% |
| | Pharmaceutical Management | 2 | 12.% |
| | Other | 2 | 12% |
| Hospital Level | Grade III, Class A | 15 | 94% |
| | Grade III, Class B | 1 | 6% |

Other: fields such as Pharmaceutical Analysis and Pharmaceutics

**Expert correlation coefficients.** As shown in Table 3, all 16 consultation questionnaires were returned, resulting in a 100% response rate. The expert authority coefficient was 0.83, indicating high expertise and reliable consultation results. The coefficient of variation for each item ranged from 0.08 to 0.60, and Kendall's coefficient of concordance was 0.198, which was statistically significant (P < 0.01).

**Table 3. Expert Activity Index, Authority Coefficient, and Opinion Coordination Level in the First Round of Consultation.**

| | Indicator | First Round |
|---|---|---|
| Expert Activity Index | Questionnaire Recovery Rate | 100% |
| | Effective Recovery Rate | 100% |
| | Opinion Proposal Rate | 38.0% |
| Expert Authority Coefficient | Cs | 0.70 |
| | Ca | 0.90 |
| | Cr | 0.83 |
| Expert Opinion Coordination Level | $\chi^2$ | 101.151 |
| | CV | 0.08 ~ 0.60 |
| | W | 0.198** |

**P < 0.01

## Delphi results

After the Delphi consultation, one item was removed from the attitude scale, and two items were modified. For the behavior scale, two items were removed, and 14 items were revised, with three new items added. The highest mean score was 4.81, and the lowest was 2.75. The highest frequency of full scores was 81.25%, and the lowest was 12.50%. The coefficient of variation ranged from 0.08 to 0.60. Detailed modifications are shown in supplementary material. The final, complete Attitudes and Behaviors of Pharmacists Involved in the Management of Patient Medication Adherence includes two subscales, Attitudes of Pharmacists Involved in the Management of Patient Medication Adherence (8-item scale) and Behaviors of Pharmacists Involved in the Management of Patient Medication Adherence (25-item scale).

## Reliability and validity testing

A total of 196 questionnaires were distributed, with 183 returned, 180 of which were valid, yielding an effective response rate of 98.36%. The demographic information of the pharmacists is shown in Table 4.

Reliability and validity analysis indicated that the Cronbach's α coefficients for the scales on pharmacists' attitudes and behaviors toward medication adherence management, as well as for the overall questionnaire, were all greater than 0.7. The KMO values were above 0.8 ($P < 0.01$), as shown in Tables 5-6.

## Phase II Results: Influencing factors analysis

**Statistical analysis results.**

**Analysis of pharmacists' demographic data on psychological contract, job burnout, and attitudes and behaviors toward patient medication adherence management.** Significant differences in behavior scores were observed between male and female pharmacists ($P < 0.05$). While no overall differences were found in job burnout scores across age groups, post-hoc analysis revealed that pharmacists aged 40–49 had higher job burnout levels compared to those aged 50–59 ($P < 0.05$). Income level differences were noted in both psychological contract and behavior scores ($P < 0.05$). Detailed results are presented in Table 7.

**Correlation between pharmacists' psychological contract, job burnout, and attitudes and behaviors toward patient medication adherence management.** Spearman's correlation analysis was conducted to examine the relationships between pharmacists' psychological contract, job burnout, and their attitudes and behaviors toward medication adherence management. Results are displayed in Table 8.

**Regression analysis of pharmacists' attitudes and behaviors toward patient medication adherence management.** Multiple linear regression analysis, using psychological contract and job burnout as independent variables, and attitudes and behaviors as dependent variables, revealed that both psychological contract and job burnout significantly predicted pharmacists' attitudes toward medication adherence management. Specifically, the regression coefficient for psychological contract was 0.147 ($P < 0.05$), and for job burnout, it was −0.140 ($P < 0.01$). For behaviors, psychological contract significantly predicted pharmacist involvement, with a regression coefficient of 0.489 ($P < 0.01$). See Table 9.

## Mediating effect of job burnout

A mediation model using psychological contract as the independent variable, job burnout as the mediator, and attitudes/behaviors as the dependent variables was tested with SPSS Process 4.1, as shown in Table 10. Results showed that psychological contract negatively affected job burnout (β = −0.2554, $P < 0.01$) and positively affected attitudes (β = 0.1824, $P < 0.05$). Job burnout negatively influenced attitudes (β = −0.2181, $P < 0.01$), and psychological contract positively impacted behaviors (β = 0.2955, $P < 0.01$).

**Table 4. Demographic Characteristics of Pharmacists Participating in the Survey.**

| | Basic situation | Pharmacist（%） |
|---|---|---|
| Gender | Male | 69（38.3%） |
| | Female | 111（61.7%） |
| Age | 20-29 | 61（33.9%） |
| | 30-39 | 83（46.1%） |
| | 40-49 | 21（11.7%） |
| | 50-59 | 15（8.3%） |
| | 60+ | 0（0%） |
| Marital Status | Single | 52（28.9%） |
| | Married | 127（70.6%） |
| | Divorced/Widowed | 1（0.5%） |
| Highest Education | Diploma | 12（6.7%） |
| | Bachelor's | 119（66.1%） |
| | Master's | 46（25.6%） |
| | Doctorate | 3（1.6%） |
| Years of Work Experience | 1-5 | 67（37.2%） |
| | 6-10 | 39（21.7%） |
| | 11-15 | 41（22.8%） |
| | 16-20 | 10（5.5%） |
| | 21-25 | 8（4.4%） |
| | 26-30 | 2（1.1%） |
| | 30+ | 13（7.3%） |
| Title | No Title | 23（12.8%） |
| | Junior Pharmacist | 6（3.3%） |
| | Pharmacist | 57（31.7%） |
| | Senior Pharmacist | 79（43.9%） |
| | Deputy Chief Pharmacist | 14（7.8%） |
| | Chief Pharmacist | 1（0.5%） |
| Employment Type | Full-Time | 92（51.1%） |
| | Contract | 68（37.8%） |
| | Part-Time | 20（11.1%） |
| Position | Inpatient Pharmacist | 38（21.1%） |
| | Outpatient Pharmacist | 70（38.9%） |
| | Intravenous Therapy Pharmacist | 4（2.2%） |
| | Clinical Pharmacist | 35（19.4%） |
| | Other | 33（18.4%） |
| Income Level (CNY, ¥) | 0~2000 | 12（6.7%） |
| | 2000~4000 | 43（23.9%） |
| | 4000~6000 | 72（40.0%） |
| | 6000+ | 53（29.4%） |
| Hospital Level | Tertiary General Hospital | 135（75.0%） |
| | Tertiary specialized hospital | 27（15.0%） |
| | Secondary General Hospital | 16（8.9%） |
| | Secondary Specialist Hospital | 2（1.1%） |
| | Primary General Hospital | 0（0%） |

**Table 5. Reliability Test Statistics of the Scale for Pharmacists' Attitudes and Behaviors in Patient Medication Adherence Management.**

|  | Number of Items | Cronbach's α |
|---|---|---|
| Attitudes | 8 | 0.796 |
| Behaviors | 25 | 0.986 |

**Table 6. KMO Measure and Bartlett's Sphericity Test Results of the Pharmacist's Attitude and Behavior Scale for Participation in Patient Medication Adherence Management.**

|  |  | Attitudes | Behaviors |
|---|---|---|---|
| KMO Sample Measure |  | 0.899 | 0.963 |
| Bartlett's Test of Sphericity Significance | Approximate Chi-Square | 1002.900 | 6275.223 |
|  | Degree of freedom | 28 | 300 |
|  | Significance probability | 0.000 | 0.000 |

**Table 7. Analysis of the scores of pharmacists' psychological contract, job burnout, attitudes, and behaviors based on demographic data[M±SD].**

|  | Group | Psychological contract | Job burnout | Attitudes | Behaviors |
|---|---|---|---|---|---|
| Gender | ①Male | 4.89±0.737 | 2.90±0.940 | 4.11±0.441 | 3.51±1.052 |
|  | ②Female | 4.79±0.632 | 2.85±0.782 | 4.06±0.598 | 3.12±1.132 |
|  | F | 0.949 | 0.157 | 0.263 | 5.254 |
|  | P | 0.331 | 0.693 | 0.609 | 0.023 |
| Age | ①20-29 | 4.88±0.719 | 2.81±0.856 | 4.07±0.583 | 3.48±1.165 |
|  | ②30-39 | 4.80±0.667 | 2.88±0.838 | 4.11±0.517 | 3.11±1.078 |
|  | ③40-49 | 4.71±0.654 | 3.19±0.915 | 3.91±0.573 | 3.43±0.947 |
|  | ④50-59 | 4.99±0.566 | 2.62±0.645 | 4.20±0.448 | 3.05±1.256 |
|  | F | 0.671 | 1.574 | 1.021 | 1.606 |
|  | P | 0.571 | 0.197 | 0.385 | 0.190 |
|  | LSD/G-H | —— | ③>④* | —— | —— |
| Marital Status | ①Single | 4.76±0.712 | 2.76±0.886 | 4.04±0.550 | 3.37±1.119 |
|  | ②Married | 4.86±0.662 | 2.90±0.186 | 4.10±0.541 | 3.22±1.118 |
|  | ③Divorced/Widowed | 5.00±0.000 | 4.55±0.000 | 3.75±0.000 | 4.00±0.000 |
|  | F | 0.380 | 2.505 | 0.452 | 0.574 |
|  | P | 0.684 | 0.085 | 0.637 | 0.564 |
| Highest Education | ①Diploma | 5.18±0.423 | 2.71±0.798 | 3.94±0.499 | 3.46±1.214 |
|  | ②Bachelor's | 4.82±0.688 | 2.92±0.879 | 4.05±0.526 | 3.21±1.121 |
|  | ③Master's | 4.77±0.758 | 2.78±0.789 | 4.19±0.593 | 3.33±1.106 |
|  | ④Doctorate | 4.79±1.068 | 2.70±0.335 | 4.21±0.402 | 3.80±0.771 |
|  | F | 1.249 | 0.504 | 1.096 | 0.493 |
|  | P | 0.294 | 0.680 | 0.352 | 0.687 |
| Years of Work Experience | ①1-5 | 4.85±0.716 | 2.79±0.788 | 4.05±0.541 | 3.33±1.150 |
|  | ②6-10 | 4.78±0.641 | 2.90±0.791 | 4.20±0.562 | 3.08±1.088 |
|  | ③11-15 | 4.85±0.625 | 2.86±0.878 | 4.05±0.567 | 3.39±1.003 |
|  | ④16-20 | 4.80±0.850 | 2.81±1.078 | 4.11±0.582 | 3.40±1.408 |
|  | ⑤21-25 | 4.79±0.799 | 3.80±1.053 | 3.78±0.499 | 3.61±1.158 |
|  | ⑥26-30 | 3.91±0.265 | 2.84±0.996 | 3.69±0.442 | 1.68±0.849 |
|  | ⑦30+ | 5.00±0.510 | 2.71±0.645 | 4.20±0.352 | 3.01±1.279 |

*(Continued)*

| | Group | Psychological contract | Job burnout | Attitudes | Behaviors |
|---|---|---|---|---|---|
| | F | 0.807 | 1.862 | 1.089 | 1.260 |
| | P | 0.565 | 0.090 | 0.371 | 0.279 |
| Title | ①No Title | 5.07±0.638 | 2.67±0.783 | 4.20±0.529 | 3.42±1.155 |
| | ②Junior Pharmacist | 5.10±0.709 | 2.42±0.789 | 4.04±0.557 | 3.19±1.198 |
| | ③Pharmacist | 4.91±0.685 | 2.83±0.948 | 4.11±0.523 | 3.43±1.226 |
| | ④Senior Pharmacist | 4.65±0.673 | 2.93±0.716 | 4.05±0.567 | 3.13±1.007 |
| | ⑤Deputy Chief Pharmacist | 4.98±0.503 | 3.25±1.074 | 3.96±0.532 | 3.10±1.187 |
| | ⑥Chief Pharmacist | 5.30±0.000 | 1.86±0.000 | 4.25±0.000 | 4.04±0.000 |
| | F | 2.304 | —— | 0.488 | 0.744 |
| | H | —— | 6.117 | —— | —— |
| | P | 0.047 | 0.289 | 0.785 | 0.592 |
| Employment Type | ①Full-Time | 4.78±0.663 | 2.97±0.845 | 4.09±0.583 | 3.32±1.072 |
| | ②Contract | 4.92±0.655 | 2.74±0.880 | 4.09±0.520 | 3.23±1.206 |
| | ③Part-Time | 4.78±0.788 | 2.83±0.671 | 4.01±0.422 | 3.12±1.019 |
| | F | 0.878 | 1.432 | 0.210 | 0.314 |
| | P | 0.417 | 0.242 | 0.811 | 0.731 |
| Position | ①Inpatient Pharmacist | 5.00±0.711 | 2.68±0.981 | 4.15±0.480 | 3.34±1.230 |
| | ②Outpatient Pharmacist | 4.80±0.626 | 2.97±0.838 | 3.98±0.547 | 3.19±1.177 |
| | ③Intravenous Therapy Pharmacist | 5.37±0.919 | 2.57±0.494 | 3.91±0.449 | 3.35±1.247 |
| | ④Clinical Pharmacist | 4.62±0.612 | 2.93±0.807 | 4.17±0.561 | 3.36±0.867 |
| | ⑤Other | 4.88±0.721 | 2.86±0.746 | 4.16±0.574 | 3.24±1.123 |
| | F | 0.172 | 0.456 | 0.330 | 0.939 |
| | P | 0.070 | 0.463 | 0.269 | 0.942 |
| Income Level | ①0-2000 | 5.26±0.842 | 2.58±1.361 | 4.30±0.428 | 3.99±1.282 |
| | ②2000-4000 | 4.82±0.778 | 2.87±0.884 | 4.00±0.616 | 3.64±1.018 |
| | ③4000-6000 | 4.74±0.625 | 2.89±0.657 | 4.07±0.520 | 3.06±1.089 |
| | ④6000+ | 4.87±0.581 | 2.91±0.904 | 4.10±0.528 | 3.08±1.071 |
| | F | 0.199 | 0.882 | 0.295 | 0.008 |
| | P | 0.090 | 0.660 | 0.394 | 0.003 |
| | LSD/G-H | —— | —— | —— | ②>③# ②>④# |
| Hospital Level | ①Tertiary General Hospital | 4.85±0.695 | 2.84±0.833 | 4.05±0.565 | 3.34±1.028 |
| | ②Tertiary specialized hospital | 4.84±0.657 | 3.23±0.916 | 4.25±0.480 | 3.31±1.406 |
| | ③Secondary General Hospital | 4.65±0.564 | 2.58±0.688 | 4.06±0.421 | 2.64±1.196 |
| | ④Secondary Specialist Hospital | 5.11±0.053 | 2.52±0.675 | 3.75±0.000 | 2.60±0.566 |
| | F | 0.001 | 0.244 | —— | 0.225 |
| | P | 0.671 | 0.064 | 0.293 | 0.094 |
| | LSD/G-H | ①<④# ③<④# | —— | ①>④# ②>④# ③>④# | —— |

*LSD; #DunnettT

**Table 8. Analysis of the Correlation between Pharmacists' Psychological Contract, Professional burnout, and Their Attitudes and Behaviors in Managing Patient Medication Adherence.**

| Project | Psychological Contract | Professional burnout | Attitudes | Behaviors |
|---|---|---|---|---|
| Psychological Contract | 1 | | | |
| Professional burnout | −0.272** | 1 | | |
| Attitudes | 0.221** | −0.265** | 1 | |
| Behaviors | 0.297** | −0.088 | 0.271** | 1 |

**P<0.01

**Table 9. The Regression Analysis of Psychological Contract, Professional burnout, and Attitudes and Behaviors.**

| Model | | Non-standardized coefficient | | standardized coefficient | t | P | VIF |
|---|---|---|---|---|---|---|---|
| | | B | standard error | β | | | |
| Attitudes | constant | 3.774 | 0.349 | | 10.820 | 0.000 | |
| | Psychological Contract | 0.147 | 0.059 | 0.182 | 2.475 | 0.014 | 1.070 |
| | Professional burnout | −0.140 | 0.047 | −0.218 | −2.959 | 0.004 | 1.070 |
| | R2 | | | | 0.101 | | |
| | F | | | | 9.961 | | |
| | P | | | | 0.000 | | |
| Behaviors | constant | 0.878 | 0.724 | | 1.213 | 0.227 | |
| | Psychological Contract | 0.489 | 0.123 | 0.296 | 3.977 | 0.000 | 1.070 |
| | Professional burnout | 0.009 | 0.098 | 0.007 | 0.097 | 0.923 | 1.070 |
| | R2 | | | | 0.086 | | |
| | F | | | | 8.360 | | |
| | P | | | | 0.000 | | |

**Table 10. Regression analysis between variables.**

| The regression equation | | The overall fit index | | | The significance of regression coefficients | |
|---|---|---|---|---|---|---|
| result variable | Predictor | R | R2 | F | β | t |
| Professional burnout | Psychological Contract | 0.2554 | 0.0652 | 12.4204 | −0.2554 | −3.5243** |
| Attitudes | Psychological Contract | 0.3181 | 0.1012 | 9.9608 | 0.1824 | 2.4753* |
| | Professional burnout | | | | −0.2181 | −2.9588** |
| Behaviors | Psychological Contract | 0.2938 | 0.0863 | 8.3597 | 0.2955 | 3.9769** |
| | Professional burnout | | | | 0.0072 | 0.0966 |

*P<0.05; **P<0.01

Further testing of the mediating effect using the non-parametric percentile Bootstrap method (Table 11) indicated that in the attitude model, the indirect effect's 95% CI did not include 0, suggesting job burnout partially mediated the relationship between pharmacists' psychological contract and attitudes toward medication adherence management (effect size = 0.0448). However, in the behavior model, the CI included 0, indicating job burnout did not mediate the relationship between psychological contract and behavior. The specific pathways are illustrated in Figs 1-2.

**Table 11. Table of Decomposition of Total Effect, Direct Effect, and Mediated Effect.**

| Model | Type of effect | Effect | standard error | t/Z | P | Boot CI Upper Limit | Boot CI lower limit | Proportion of effect |
|---|---|---|---|---|---|---|---|---|
| Attitudes | Total | 0.1914 | 0.0585 | 3.2713 | 0.0013 | 0.0759 | 0.3068 | 100.00% |
| | Direct | 0.1466 | 0.0592 | 2.4753 | 0.0143 | 0.0297 | 0.2635 | 76.59% |
| | Professional burnout | 0.0448 | 0.0203 | | | 0.0113 | 0.0899 | 23.51% |
| Behaviors | Total | 0.4858 | 0.1185 | 4.0992 | 0.0001 | 0.2519 | 0.7197 | 100.00% |
| | Direct | 0.4888 | 0.1229 | 3.9769 | 0.0001 | 0.2463 | 0.7314 | 100.62% |
| | Professional burnout | −0.0003 | 0.0381 | | | −0.0823 | 0.0724 | −0.62% |

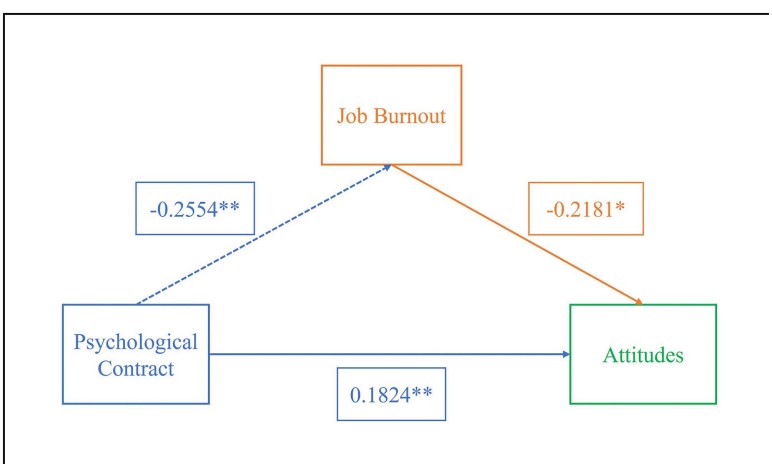

**Fig 1. The Mediating Model of Pharmacist Burnout Between Psychological Contract and Attitude Toward Patient Medication Adherence Management.**

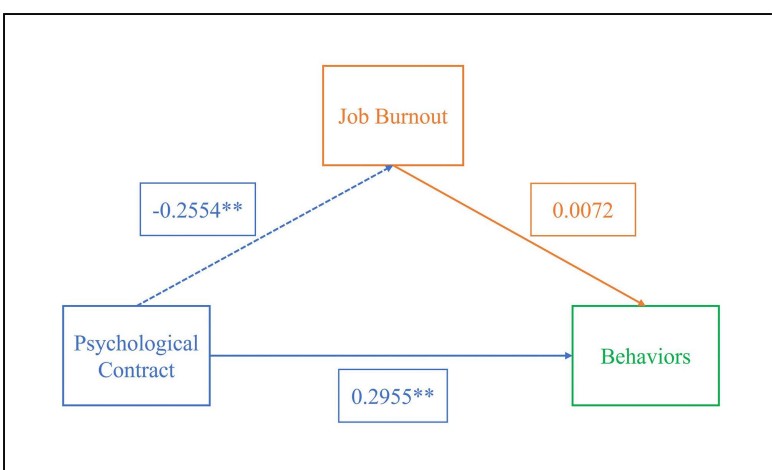

**Fig 2. The Mediating Model of Pharmacist Burnout Between Psychological Contract and Behavior Toward Patient Medication Adherence Management.**

## Discussion

Medication adherence is widely recognized as a pivotal determinant of clinical treatment effectiveness, yet remains influenced by a multifactorial set of patient-related, therapy-related, and system-level factors [26]. Recent studies have emphasized the growing importance of healthcare professionals, particularly pharmacists, in mitigating non-adherence through patient-centered interventions [27]. In this context, pharmacists' engagement and motivation are increasingly acknowledged as critical components for ensuring medication adherence outcomes [28].

This study highlights the relationship between pharmacists' psychological contracts and their attitudes and behaviors in medication adherence management. These findings align with our previous research on the connection between pharmacists' psychological contracts and their attitudes and behaviors in pharmacy services [10]. Specifically, when pharmacists perceive a higher level of psychological contract fulfillment, they demonstrate more positive and proactive attitudes and behaviors toward managing patient medication adherence. In turn, this commitment enhances their relationship with the hospital, fosters patient trust, and promotes better treatment outcomes, all of which contribute to a stronger doctor-patient relationship [29]. A growing body of evidence indicates that pharmacists' service attitudes—encompassing patient-centred communication, empathy, responsiveness, and clarity of counselling—shape patients' trust in pharmacists, which subsequently serves as a critical determinant of medication adherence [30]. Consistent with psychological contract theory, pharmacists who perceive their implicit agreements with organisations as fulfilled—across dimensions of fairness, relational respect, and developmental opportunities—tend to regard adherence management as a core professional obligation and engage more proactively in these activities [31]. Contract fulfilment strengthens role identity and self-efficacy, promoting behaviours such as adherence monitoring, patient outreach, and collaboration with prescribers, whereas breaches diminish willingness to invest discretionary effort and may exacerbate burnout. Therefore, clarifying role expectations, actively fulfilling the psychological contract with pharmacists, and implementing organizational strategies that provide structured support are essential for sustaining pharmacists' engagement in medication adherence management. These measures not only reinforce pharmacists' sense of professional identity and responsibility but also foster a supportive work environment that reduces ambiguity and stress, ultimately promoting consistent and high-quality participation in adherence-related interventions.

Pharmacists with a robust sense of psychological contract recognize the importance of adherence management in improving treatment outcomes, reducing healthcare costs, and enhancing patients' quality of life. As a result, they are actively involved in this area. However, when the psychological contract is breached, pharmacists may feel disillusioned, leading to a decline in their motivation to manage medication adherence. This lack of motivation can result in diminished work quality, causing patients to question the efficacy and safety of their treatment regimens, which may, in turn, reduce medication adherence. Thus, fulfilling the psychological contract between healthcare institutions and pharmacists is crucial for encouraging their active participation in medication adherence management. Healthcare organizations should prioritize fulfilling these contracts by improving communication, ensuring fair workload distribution, and offering career development support to enhance pharmacists' perception of their psychological contract. This approach will foster pharmacists' enthusiasm and initiative in managing medication adherence, ultimately improving patient adherence and strengthening the doctor-patient relationship.

Additionally, this study identified burnout as a critical mediating variable linking the psychological contract to pharmacists' participation in medication adherence management. Our findings revealed that burnout significantly mediated this relationship, demonstrating a negative association with both psychological contract fulfillment and attitudes toward adherence management. The psychological contract serves as an implicit understanding between pharmacists and healthcare organizations, shaping expectations of mutual obligations. When pharmacists perceive that this contract is honored—through recognition, respect, and fair support—they are more likely to experience a sense of professional value and meaning, which enhances job satisfaction and mitigates burnout [32]. In contrast, perceived breaches in the psychological contract, such as inadequate recognition, limited growth opportunities, or an imbalance between effort and reward, can

heighten emotional exhaustion and cynicism, thereby accelerating burnout. Elevated burnout, in turn, erodes pharmacists' motivation and engagement, diminishing their willingness to invest effort in adherence-related interventions and ultimately compromising the quality of patient support and treatment outcomes [33].

High levels of burnout can cause pharmacists to lose enthusiasm and motivation, diminishing their attention to medication adherence management. They may become less proactive in communicating with patients or providing necessary medication guidance and follow-up. For example, they may adopt a perfunctory attitude, offering brief and impatient responses to patients' inquiries. Emotional distress may also impair their communication with patients, reducing trust and, consequently, medication adherence [34]. Therefore, hospitals and policymakers must prioritize addressing pharmacist burnout. This can be achieved by enhancing pharmacists' perceptions of their psychological contract through respect, support, and professional development opportunities, as well as optimizing the work environment to reduce negative factors contributing to burnout. Additionally, implementing regular burnout screening as part of surveillance and quality measurement initiatives could help identify early warning signs and areas for improvement. These findings can then inform targeted quality improvement interventions. Such measures will help prevent a decline in work quality and improve medication adherence management.. Collectively, these findings contribute to a growing body of research advocating for the integration of psychological and organizational perspectives into adherence improvement frameworks. Future studies should consider longitudinal designs to explore the dynamic interplay between psychological contract evolution, burnout trajectories, and adherence-related behaviors over time, particularly within diverse healthcare settings and cultural contexts

This study aims to explore the psychological factors influencing pharmacists' participation in medication adherence management and examine the mediating role of burnout. It provides empirical insights into the psychological mechanisms underlying pharmacists' involvement in medication adherence management, offering theoretical support and practical guidance for healthcare organizations to develop more targeted management strategies. Based on these findings, the following recommendations are made to optimize pharmacists' motivation in medication adherence management and improve patient adherence: (1) Enhance the fulfillment of pharmacists' psychological contracts, such as strengthening trust between pharmacists and healthcare institutions, and improving communication between hospital management and pharmacists. Enhance the Fulfillment of Pharmacists' Psychological Contracts: Strengthen trust between pharmacists and healthcare institutions, improve communication between hospital management and pharmacists, and ensure psychological contract fulfillment. (2) Reduce burnout by improving working conditions, providing career development opportunities, and prioritizing pharmacists' psychological well-being. Reduce Burnout: Improve working conditions, offer career development opportunities, and prioritize pharmacists' psychological well-being to reduce burnout and enhance job satisfaction. (3) Boost pharmacists' motivation to participate in medication adherence management, e.g., fostering a sense of responsibility through training, establishing incentive mechanisms (e.g., incorporating participation in medication adherence management into performance evaluations and offering appropriate rewards), and promoting teamwork (e.g., enhancing collaboration between pharmacists and other healthcare professionals to provide comprehensive medication management services and improve adherence management outcomes). Boost Pharmacists' Motivation: Foster a sense of responsibility through training, establish incentive mechanisms (e.g., incorporating medication adherence management into performance evaluations and offering appropriate rewards), and promote teamwork (e.g., enhancing collaboration between pharmacists and other healthcare professionals to provide comprehensive medication management and improve adherence outcomes).

## Limitations

This study has several limitations that warrant further investigation. First, only one round of the Delphi survey was conducted with a limited number of experts, which may impact the depth and comprehensiveness of the findings. Future research should involve multiple rounds of the Delphi survey and expand the expert panel to incorporate a broader

range of perspectives, thereby enhancing the diversity and representativeness of the results. Second, the study focused exclusively on pharmacists in hospitals in Zunyi and Bijie, Guizhou Province, China, limiting the geographic scope of the research. Expanding the study to include regions with varying levels of economic and healthcare resources would improve the generalizability of the findings. Third, while this study explored burnout as a mediating variable in the relationship between pharmacists' psychological contracts and their involvement in medication adherence management, it may not fully capture the underlying mechanisms of pharmacists' behaviors. Future research should incorporate additional variables, such as professional responsibility and identity, to develop a more comprehensive theoretical model and enhance the explanatory power of the relationship between pharmacists' psychological contracts and engagement in medication adherence management. Lastly, this study primarily relied on quantitative surveys, which may not fully capture the complex experiences and psychological factors influencing pharmacists in real-world settings. Future research should integrate qualitative methods, such as in-depth interviews and case studies, with quantitative data to provide a more comprehensive analysis of the challenges faced by hospital pharmacists, offering deeper insights to inform decision-making.

## Conclusions

The fulfillment of pharmacists' psychological contracts significantly influences their engagement in medication adherence management, with job burnout acting as a mediating factor. Furthermore, the psychological contract is a key determinant of pharmacist burnout. Effective management of pharmacists' psychological contracts is thus crucial for enhancing motivation, reducing burnout, improving patient adherence, and optimizing medication safety and treatment outcomes. The findings of this study offer an empirical foundation for understanding the factors that shape pharmacists' attitudes and behaviors in medication adherence management. They also provide theoretical support and practical guidance for healthcare organizations in developing strategies to manage pharmacists' psychological contracts. However, future studies should broaden the scope by exploring additional mediating variables, such as professional responsibility and identity, and investigating different types of pharmacists (e.g., community pharmacists, clinical pharmacists) and patient populations (e.g., chronically ill or elderly patients) to examine potential variations in the psychological contract's role across diverse groups. Moreover, integrating qualitative research alongside quantitative methods could offer a more nuanced understanding of the psychological factors that influence pharmacists' workplace behaviors, thus enriching our understanding of the dynamic relationship between psychological contracts and medication adherence management.

## Abbreviations

The following abbreviations are used in this manuscript:

| | |
|---|---|
| Cr | Authority Coefficient |
| Cs | Coefficient of Familiarity |
| Ca | Coefficient of Judgment Basis |
| CV | Coefficient of Variation |
| KMO | Kaiser-Meyer-Olkin |

## Supporting information

**S1 File. This supporting information contains the full survey instrument used in this study and the feedback received from the Delphi experts.**
(DOCX)

## Acknowledgments

We thank all the pharmacists who actively participated in this study.

## Author contributions

**Conceptualization:** Fushan Tang.

**Data curation:** Yongyong Luo, Yang Gu.

**Formal analysis:** Yongyong Luo, Xiaoyu Jiang, Ting Zhang.

**Investigation:** Yongyong Luo, Mei Nie, Cheng Chen.

**Methodology:** Yongyong Luo, Mei Nie.

**Project administration:** Fushan Tang.

**Validation:** Yong He, Jianhua Tang.

**Visualization:** Fushan Tang.

**Writing – original draft:** Yongyong Luo.

**Writing – review & editing:** Jianhua Tang, Fushan Tang.

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
