## [Decision Letter · Decision Letter 0]

23 Jul 2025

Dear Dr. Tang,

Thank you for submitting your manuscript to PLOS ONE. After careful consideration, we feel that it has merit but does not fully meet PLOS ONE’s publication criteria as it currently stands. Therefore, we invite you to submit a revised version of the manuscript that addresses the points raised during the review process.

We look forward to receiving your revised manuscript.

Kind regards,

Ali Ahmed, PhD

Academic Editor

PLOS ONE

Journal Requirements:

This study was supported by the Special Funds for Science and Technology Cooperation in Guizhou Province and Zunyi City (Shengshikehe 2015] 53) and the Graduate Student Research Fund of Zunyi Medical University (ZYK233).

4. In the online submission form, you indicated that the datasets used and analyzed in this study can be obtained from the corresponding author upon reasonable request.

Reviewers' comments:

Reviewer's Responses to Questions

**Comments to the Author**

1. Is the manuscript technically sound, and do the data support the conclusions?

Reviewer #1: Partly

Reviewer #2: Yes

2. Has the statistical analysis been performed appropriately and rigorously?

Reviewer #1: No

Reviewer #2: I Don't Know

3. Have the authors made all data underlying the findings in their manuscript fully available?

Reviewer #1: Yes

Reviewer #2: Yes

4. Is the manuscript presented in an intelligible fashion and written in standard English?

Reviewer #1: Yes

Reviewer #2: Yes

Reviewer #1: An interesting study has been conducted. Medication Adherence is one of the most important concepts in treating patients. The manuscript was reviewed.

There are some concerns for me as I write:

In the Introduction, it is not clear on what basis the correlation between the variables Psychological Contracts, Medication1

Adherence and Job Burnout was measured. Was this based on a model or theory or was it based on a literature review?

The method is ambiguous. For example, it is written that A comprehensive literature search was conducted to prepare a draft scale "Pharmacist's Attitudes and Behaviors Toward Patient

Medication Adherence Management". The literature search was also a systematic review study. However, there is no information about the method of this systematic review study. For example, the number of articles reviewed, the number of articles excluded, the eligibility of articles, etc. It would have been better to add a prism diagram.

How was the item pool formed in the preparation of this scale and how many items did it have?

What steps were taken in the next step, Delphi Expert Consultation?

Expert Evaluation and Data Analysis was based on a formula. What is the reference of this formula? And how is it evaluated? In what range is accepted?

In the Participant Recruitment stage, what was the sample size and sampling method? How was the validity of the scale determined?

On what basis was the Expert's judgment made?

The tables provided are too many. 12 tables!

The discussion, considering the results of other studies, needs a general review.

Reviewer #2: Thank you for sharing this work for review.

I have a few suggestions for improvement:

1. (line 99) in Method section related to detail on 'participant recruitment', please mention if any incentives were given or not.

2. (line 126) in Results, in table 2, add explanation to 'research field' showing 2 participants as 'others', which were these?

3. (line 144) in table 5, please mention the unit of currency for 'income level'.

4. (line 239) in discussion, add that burnout screening should be done quarter-wise or biannually, for surveillance or quality measure, to identify areas of improvement through targeted quality improvement initiatives.

.

Reviewer #1: No

Reviewer #2: **Yes:**Adeel SiddiquiAdeel SiddiquiAdeel SiddiquiAdeel Siddiqui

---

## [Author Response · Author response to Decision Letter 1]

21 Sep 2025

Reviewer 1

Comments 1: In the Introduction, it is not clear on what basis the correlation between the variables Psychological Contracts, Medication Adherence and Job Burnout was measured. Was this based on a model or theory or was it based on a literature review?

Response 1: Thank you for your valuable comment. We agree that the theoretical basis for the proposed relationships needed to be clarified. In the revised Introduction, we have added a paragraph to explain that our research framework is grounded in psychological contract theory and social exchange theory. We highlight that fulfillment of psychological contracts enhances pharmacists’ motivation and professional engagement, whereas breaches may foster job burnout, which in turn undermines pharmacists’ behaviors in supporting patients’ medication adherence. These links are supported by prior studies on healthcare professionals, providing a sound theoretical and empirical basis for our research model. Please see lines 59–66 of the revised manuscript.

Comments 2: The method is ambiguous. For example, it is written that A comprehensive literature search was conducted to prepare a draft scale "Pharmacist's Attitudes and Behaviors Toward Patient Medication Adherence Management". The literature search was also a systematic review study. However, there is no information about the method of this systematic review study. For example, the number of articles reviewed, the number of articles excluded, the eligibility of articles, etc. It would have been better to add a prism diagram.

Response 2: Thank you for your insightful comment. We acknowledge that our description may have caused confusion. Our literature search was intended as an exploratory review to inform the development of the scale, rather than a formal systematic review. Therefore, we did not apply PRISMA methodology or report article inclusion/exclusion numbers. To avoid misunderstanding, we have revised the Methods section to clarify that the literature search was exploratory in nature and primarily aimed to generate an item pool, which was then refined through expert consultations.

Comments 3: How was the item pool formed in the preparation of this scale and how many items did it have? What steps were taken in the next step, Delphi Expert Consultation?

Response 3: Thank you very much for your insightful comments. Regarding your question:

(1) Formation of the item pool

The item pool was developed through a targeted literature review rather than a formal systematic review. We searched both Chinese (CNKI, Wanfang, VIP) and international databases (PubMed, Web of Science) using keywords such as "Medication Adherence," "Pharmacist," "Psychological Contract," and "Management." Relevant literature, policy documents, and practice guidelines were synthesized to identify common domains of pharmacists’ attitudes and behaviors related to patient medication adherence management. Based on this process, an initial item pool was generated, including 9 attitude items and 24 behavior items, for a total of 33 items.

(2) Delphi expert consultation

The Delphi process was conducted in a single round with 16 experts. Experts rated the importance and clarity of each item and provided qualitative feedback. Based on their suggestions, redundant or ambiguous items were revised or deleted, and consensus was reached on the final set of items. This process ensured content validity and expert agreement regarding the scale structure.

Comments 4: Expert Evaluation and Data Analysis was based on a formula. What is the reference of this formula? And how is it evaluated? In what range is accepted?

Response 4: Thank you for your valuable comment. The formula used for calculating the expert authority coefficient (Cr) is based on standard methodology commonly applied in Delphi studies in health services research. Specifically, Cr is calculated as (Cs + Ca)/2, where Cs represents the expert’s familiarity with the questionnaire and Ca reflects the basis of the expert’s judgment. This approach has been adopted in previous studies (Dai F, Wei K, Chen Y, Ju M. Construction of an index system for qualitative evaluation of undergraduate nursing students innovative ability: A Delphi study. J Clin Nurs. 2019;28(23-24):4379-4388. doi:10.1111/jocn.15020).

In our study, a Cr value ≥ 0.70 was considered to indicate good reliability, consistent with the thresholds reported in earlier methodological research (Hu Q, Qin Z, Zhan M, Wu B, Chen Z, Xu T. Development of a trigger tool for the detection of adverse drug events in Chinese geriatric inpatients using the Delphi method. Int J Clin Pharm. 2019;41(5):1174-1183. doi:10.1007/s11096-019-00871-x). The ranges for Cs (0.1–0.9) and Ca (0.1–0.5, depending on the strength of the judgment basis) are presented in Table 1 of our manuscript.

Regarding evaluation, expert authority was judged as acceptable when Cr ≥ 0.70, expert opinion concentration was confirmed by an average importance score ≥ 3.5, and coordination of expert opinions was verified with CV < 0.30 and Kendall’s W showing P < 0.05, following established Delphi methodology (de Goumoëns V, Lefrançois LE, Forestier A, et al. Bachelor nursing competencies to care for children in hospital and home settings: A Delphi study. Nurse Educ Today. 2025;145:106487. doi:10.1016/j.nedt.2024.106487).

Comments 5: In the Participant Recruitment stage, what was the sample size and sampling method? How was the validity of the scale determined?

Response 5: We appreciate the reviewer’s insightful comment. In the Participant Recruitment stage, a stratified random cluster sampling method was adopted to recruit participants from tertiary, secondary, and primary healthcare institutions in Zunyi and Bijie, China. Between November and December 2024, a total of 196 valid responses were obtained.

Regarding scale validity, the Kaiser–Meyer–Olkin (KMO) measure and Bartlett’s test of sphericity were employed to evaluate sampling adequacy and the suitability of the data for factor analysis. A KMO value above 0.7 together with a significant Bartlett’s test result (P < 0.05) confirmed that the data were appropriate for further analysis, thereby supporting the structural validity of the scale in line with the theoretical framework.

Comments 6: On what basis was the Expert's judgment made?

Response 6: The experts’ judgments were based on their professional background and extensive experience in hospital pharmacy and clinical practice. All invited experts held senior professional titles (associate chief pharmacist or chief pharmacist) and had more than ten years of work experience. They evaluated each item in terms of its importance, clarity, and relevance to pharmacists’ involvement in patient medication adherence management, and provided qualitative feedback. This ensured that the scale items reflected both theoretical rigor and practical applicability from the perspective of frontline pharmacists.

Comments 7: The tables provided are too many. 12 tables!

Response 7: We sincerely appreciate the reviewer’s suggestion regarding the number of tables. We fully understand the concern; however, the tables included in the manuscript serve distinct and essential purposes. Specifically, they present (1) the Delphi process results, (2) the reliability and validity testing of the scale, (3) the analysis of the relationship between pharmacists’ psychological contract and medication adherence management, and (4) the mediation analysis of job burnout. Each table corresponds to a separate stage of data processing or statistical analysis, and merging or removing them may compromise the clarity and completeness of the findings. For this reason, we respectfully maintain all tables to ensure transparency and scientific rigor in presenting the study results.

Comments 8: The discussion, considering the results of other studies, needs a general review.

Response 8: Thank you for your valuable comment. In response, we have thoroughly revised the Discussion section to provide a more comprehensive and balanced review of the literature. Specifically, we have incorporated comparative analyses, highlighting both consistencies with previous studies and observed discrepancies, along with potential explanations. We have emphasized the novelty of our study, particularly how it addresses gaps in the literature regarding hospital pharmacists’ roles in medication adherence management and the mediating effect of burnout. Additionally, the logical structure of the Discussion has been strengthened, following a clear sequence: major findings, consistencies with prior research, observed differences and possible explanations, study significance and innovations, and limitations with directions for future research. These revisions ensure that the Discussion is both comprehensive and well contextualized within the current body of evidence. Please see lines 214–271 of the revised manuscript.

Reviewer 2

Comments 1: (line 99) in Method section related to detail on 'participant recruitment', please mention if any incentives were given or not.

Response 1: We appreciate the reviewer’s valuable comment. We have now clarified in the Methods – Participant recruitment section that no financial or material incentives were provided for participation. The revised sentence reads: “No financial or material incentives were provided for participation”.

Comments 2: (line 126) in Results, in table 2, add explanation to 'research field' showing 2 participants as 'others', which were these?

Response 2: Thank you for your careful observation. In our questionnaire, the category “Other” was predefined to capture participants whose research fields did not fit within Clinical Pharmacy, Hospital Pharmacy, or Pharmaceutical Management. The two experts classified under “Other” were engaged in areas such as Pharmaceutical Analysis, Pharmaceutics, and other pharmacy-related disciplines. Although these fields play important roles within hospital pharmacy practice, their direct contribution to pharmaceutical management research is comparatively limited, which is why they were categorized as “Other.” This approach ensured inclusivity and avoided misclassification of participants’ expertise. We have now clarified this point in the Results section (Table 2 note: Other: fields such as Pharmaceutical Analysis and Pharmaceutics).

Comments 3: (line 144) in table 5, please mention the unit of currency for 'income level'.

Response 3: Thank you for your careful observation. The income levels reported in Table 5 are expressed in Chinese Yuan (CNY, ¥). We have added this clarification to the table note.

Comments 4: (line 239) in discussion, add that burnout screening should be done quarter-wise or biannually, for surveillance or quality measure, to identify areas of improvement through targeted quality improvement initiatives.

Response 4: Thank you for your valuable suggestion. We agree that regular burnout screening is essential for early detection and continuous quality improvement. Accordingly, we have revised the discussion to include the recommendation that hospitals conduct burnout screening on a quarterly or biannual basis as part of surveillance and quality measurement strategies, enabling timely interventions and targeted improvements (see Discussion, lines 262–271).

---

## [Decision Letter · Decision Letter 1]

8 Dec 2025

Dear Dr. Tang,

Thank you for submitting your manuscript to PLOS ONE. After careful consideration, we feel that it has merit but does not fully meet PLOS ONE’s publication criteria as it currently stands. Therefore, we invite you to submit a revised version of the manuscript that addresses the points raised during the review process.

An interesting study has been conducted. Medication Adherence is one of the most important concepts in treating patients. The manuscript was reviewed. There are some concerns for me as I write:In the Introduction, it is not clear on what basis the correlation between the variables Psychological Contracts, Medication1 Adherence and Job Burnout was measured. Was this based on a model or theory or was it based on a literature review?The method is ambiguous. For example, it is written that A comprehensive literature search was conducted to prepare a draft scale "Pharmacist's Attitudes and Behaviors Toward Patient Medication Adherence Management". The literature search was also a systematic review study. However, there is no information about the method of this systematic review study.For example, the number of articles reviewed, the number of articles excluded, the eligibility of articles, etc. It would have been better to add a prism diagram.How was the item pool formed in the preparation of this scale and how many items did it have? What steps were taken in the next step, Delphi Expert Consultation?Expert Evaluation and Data Analysis was based on a formula.What is the reference of this formula? And how is it evaluated?In what range is accepted?In the Participant Recruitment stage, what was the sample size and sampling method? How was the validity of the scale determined? On what basis was the Expert's judgment made?The tables provided are too many. 12 tables!The discussion, considering the results of other studies, needs a general review. Reviewer 2 commentsThis research is interesting, however, the presentation is a bit confusing. I suggest that the authors reformate it in a more systematic ways. So, this research contains two stages of research.The first one is the development and validation of the medication adherence management attitude and behaviour scale and the second one is using that scale to measure factors influencing it (psychological contract and job burn out).The authors should systematically write it as two stages in the methods and explained it clearly the methods used for each stages. The combination of this two stages of research has make presentation of the results become too much with all 12 tables, with some of them are very big tables. This was not effective for an article. This has also makes the explanation about the important variables measured and other important part of methods were missing. One example include there is no information about the instrument used for psychological contract and job burn out, and what it is actually about.The authors has mentioned about the theories on the psychological contract and job burnout with work outcomes. But has not been strong enough why part of the work outcomes is their attitude and behaviour in medication adherence management and not the other pharmacist job?The authors need to provide the context and the problem for this for example whether this job is a mandatory task for pharmacist in hospitals in the study location, and what the problem about this specific task. The above are the comments of the two reviewers. Draft a letter based on the above comments to the author to rectify and address.

We look forward to receiving your revised manuscript.

Kind regards,

Asim Mehmood

Academic Editor

PLOS One

Journal Requirements:

Additional Editor Comments:

Dear Authors,

Thank you for submitting your manuscript. While both reviewers found the topic important and relevant—particularly regarding medication adherence—they identified several areas that require major clarification and restructuring before the manuscript can be reconsidered.

Please revise the manuscript by addressing the points outlined below.

✔ Clarify the theoretical framework behind the selected variables

✔ Provide full systematic review methodology with a PRISMA diagram

✔ Clearly describe scale development, item pool, and Delphi process

✔ Add references and interpretation for the expert evaluation formula

✔ Provide complete details on sample recruitment and scale validation

✔ Describe instruments for psychological contract and job burnout

✔ Strengthen contextual justification for choosing medication adherence management

✔ Reorganize the Methods and Results into two clear stages

✔ Reduce the number of tables

✔ Revise and strengthen the Discussion section

Reviewers' comments:

Reviewer's Responses to Questions

**Comments to the Author**

Reviewer #2: All comments have been addressed

Reviewer #3: (No Response)

2. Is the manuscript technically sound, and do the data support the conclusions?

Reviewer #2: Yes

Reviewer #3: Partly

3. Has the statistical analysis been performed appropriately and rigorously?

Reviewer #2: Yes

Reviewer #3: Yes

4. Have the authors made all data underlying the findings in their manuscript fully available?

Reviewer #2: Yes

Reviewer #3: Yes

5. Is the manuscript presented in an intelligible fashion and written in standard English?

Reviewer #2: Yes

Reviewer #3: No

Reviewer #2: As the comments are rightly addressed, I recommend acceptance of this manuscript. This work will add significant knowledge to the existing literature.

Reviewer #3: This research is interesting, however, the presentation is a bit confusing.

I suggest that the authors reformate it in a more systematic ways. So, this research contains two stages of research. The first one is the development and validation of the medication adherence management attitude and behaviour scale and the second one is using that scale to measure factors influencing it (psychological contract and job burn out).

The authors should systematically write it as two stages in the methods and explained it clearly the methods used for each stages. The combination of this two stages of research has make presentation of the results become too much with all 12 tables, with some of them are very big tables. This was not effective for an article. This has also makes the explanation about the important variables measured and other important part of methods were missing. One example include there is no information about the instrument used for psychological contract and job burn out, and what it is actually about.

The authors has mentioned about the theories on the psychological contract and job burnout with work outcomes. But has not been strong enough why part of the work outcomes is their attitude and behaviour in medication adherence management and not the other pharmacist job? The authors need to provide the context and the problem for this for example whether this job is a mandatory task for pharmacist in hospitals in the study location, and what the problem about this specific task.

.

Reviewer #2: **Yes:** Adeel SiddiquiAdeel SiddiquiAdeel SiddiquiAdeel Siddiqui

Reviewer #3: No

---

## [Author Response · Author response to Decision Letter 2]

19 Jan 2026

Comments 1: In the Introduction, it is not clear on what basis the correlation between the variables Psychological Contracts, Medication Adherence and Job Burnout was measured. Was this based on a model or theory or was it based on a literature review?

Response 1: Thank you for your valuable comments. The proposed relationships among psychological contract, medication adherence management, and job burnout in this study are not based on exploratory analyses or ad hoc assumptions, but are grounded in established theoretical frameworks and supported by existing literature.

Specifically, this study is primarily informed by social exchange theory and conservation of resources theory (Dee J, Dhuhaibawi N, Hayden JC. Int J Clin Pharm. 2023;45(5):1027–1036). According to social exchange theory, employees’ perceptions of psychological contract fulfillment or breach shape their evaluations of organizational support and reciprocal obligations. Conservation of resources theory further posits that psychological contract breaches lead to ongoing depletion of individuals’ psychological resources, thereby increasing job burnout. Prior research has consistently identified job burnout as a key mediating mechanism through which psychological contracts influence work attitudes and outcomes.

In addition, existing literature indicates that healthcare professionals’ work engagement and well-being are closely associated with their performance in patient-centered professional behaviors, including medication adherence management (Ring M, Hult M. J Adv Nurs. 2025;81(3):1323–1331). Fulfillment of the psychological contract enhances healthcare professionals’ motivation and sense of responsibility, whereas elevated burnout may undermine their engagement in patient management and service delivery.

Therefore, the hypothesized relationships among the study variables were developed under these theoretical frameworks and in alignment with prior empirical evidence, providing a clear theoretical and literature-based foundation for the study. Please see lines 62–70 of the revised manuscript.

Comments 2: The method is ambiguous. For example, it is written that A comprehensive literature search was conducted to prepare a draft scale "Pharmacist's Attitudes and Behaviors Toward Patient Medication Adherence Management". The literature search was also a systematic review study. However, there is no information about the method of this systematic review study. For example, the number of articles reviewed, the number of articles excluded, the eligibility of articles, etc. It would have been better to add a prism diagram.

Response 2: Thank you for your insightful comments. We acknowledge that the description of the literature review may have caused confusion. The purpose of the literature search in this study was to provide an exploratory review to support scale development, rather than to conduct a formal systematic review. Accordingly, PRISMA methodology was not applied, and the numbers of included and excluded articles were not reported.

To avoid misunderstanding, we have clarified this point in the Methods section by explicitly stating: “The literature review had an exploratory nature, with the primary aim of developing an initial item pool, which was subsequently refined and optimized through expert consultation” (lines 94–95).

Comments 3: How was the item pool formed in the preparation of this scale and how many items did it have? What steps were taken in the next step, Delphi Expert Consultation?

Response 3: Thank you very much for your insightful comments. Regarding your question:

(1) Formation of the item pool

The item pool was developed through a targeted literature review rather than a formal systematic review. We searched both Chinese (CNKI, Wanfang, VIP) and international databases (PubMed, Web of Science) using keywords such as "Medication Adherence," "Pharmacist," "Psychological Contract," and "Management." Relevant literature, policy documents, and practice guidelines were synthesized to identify common domains of pharmacists’attitudes and behaviors related to patient medication adherence management. Based on this process, an initial item pool was generated, including 9 attitude items and 24 behavior items, for a total of 33 items.

(2) Delphi expert consultation

The Delphi process was conducted in a single round with 16 experts. Experts rated the importance and clarity of each item and provided qualitative feedback. Based on their suggestions, redundant or ambiguous items were revised or deleted, and consensus was reached on the final set of items. This process ensured content validity and expert agreement regarding the scale structure.

Comments 4: Expert Evaluation and Data Analysis was based on a formula. What is the reference of this formula? And how is it evaluated? In what range is accepted?

Response 4: Thank you for your valuable comment. The formula used for calculating the expert authority coefficient (Cr) is based on standard methodology commonly applied in Delphi studies in health services research. Specifically, Cr is calculated as (Cs + Ca)/2, where Cs represents the expert’s familiarity with the questionnaire and Ca reflects the basis of the expert’s judgment. This approach has been adopted in previous studies (Dai F, Wei K, Chen Y, Ju M. Construction of an index system for qualitative evaluation of undergraduate nursing students innovative ability: A Delphi study. J Clin Nurs. 2019;28(23-24):4379-4388. doi:10.1111/jocn.15020).

In our study, a Cr value ≥ 0.70 was considered to indicate good reliability, consistent with the thresholds reported in earlier methodological research (Hu Q, Qin Z, Zhan M, Wu B, Chen Z, Xu T. Development of a trigger tool for the detection of adverse drug events in Chinese geriatric inpatients using the Delphi method. Int J Clin Pharm. 2019;41(5):1174-1183. doi:10.1007/s11096-019-00871-x). The ranges for Cs (0.1–0.9) and Ca (0.1–0.5, depending on the strength of the judgment basis) are presented in Table 1 of our manuscript.

Regarding evaluation, expert authority was judged as acceptable when Cr ≥ 0.70, expert opinion concentration was confirmed by an average importance score ≥ 3.5, and coordination of expert opinions was verified with CV < 0.30 and Kendall’s W showing P < 0.05, following established Delphi methodology (de Goumoëns V, Lefrançois LE, Forestier A, et al. Bachelor nursing competencies to care for children in hospital and home settings: A Delphi study. Nurse Educ Today. 2025;145:106487. doi:10.1016/j.nedt.2024.106487).

Comments 5: In the Participant Recruitment stage, what was the sample size and sampling method? How was the validity of the scale determined? On what basis was the Expert's judgment made?

Response 5: We appreciate your insightful comments. During participant recruitment, a stratified random cluster sampling method was employed to recruit participants from tertiary, secondary, and primary healthcare institutions in the Zunyi and Bijie regions of China. A total of 196 valid questionnaires were collected between November and December 2024.

Regarding scale validity, sampling adequacy and the suitability of the data for factor analysis were assessed using the Kaiser–Meyer–Olkin (KMO) measure and Bartlett’s test of sphericity. KMO values greater than 0.7 and statistically significant Bartlett’s test results (P < 0.05) indicated that the data were appropriate for factor analysis, thereby supporting the construct validity of the scale in accordance with its theoretical framework.

Expert judgment was primarily based on both the experts’ judgment criteria and quantitative evaluation results. Specifically, experts’ judgment criteria included practical experience, theoretical analysis, relevant literature, and personal judgment. These criteria were weighted according to established methods in prior studies and were used to calculate the expert authority coefficient (Cr). In addition, experts rated the importance of each item using a 5-point Likert scale. The mean importance score, coefficient of variation (CV), and Kendall’s coefficient of concordance (W) were jointly used to assess the concentration and consistency of expert opinions. Ultimately, decisions regarding item retention were made by integrating quantitative indicators with experts’ qualitative feedback.

Comments 6: The tables provided are too many. 12 tables!

Response 6: We sincerely appreciate the your suggestion regarding the number of tables. We fully understand the concern; however, the tables included in the manuscript serve distinct and essential purposes. Specifically, they present (1) the Delphi process results, (2) the reliability and validity testing of the scale, (3) the analysis of the relationship between pharmacists’ psychological contract and medication adherence management, and (4) the mediation analysis of job burnout. Each table corresponds to a separate stage of data processing or statistical analysis, and merging or removing them may compromise the clarity and completeness of the findings. For this reason, we respectfully maintain all tables to ensure transparency and scientific rigor in presenting the study results.

Comments 7: The discussion, considering the results of other studies, needs a general review.

Response 7: We thank for your additional valuable comments. In response to the suggestion that the Discussion should provide a more comprehensive synthesis of relevant studies, we have systematically restructured and substantially expanded the Discussion section in the previous round of revisions.

Specifically, the revised Discussion now provides an integrated evaluation of our findings in relation to existing literature from the following perspectives:

(1) Drawing on prior studies, we systematically discuss the role of pharmacists in medication adherence management and compare our findings with those of related research, highlighting both consistencies and discrepancies;

(2) Guided by psychological contract theory and existing empirical evidence, we further analyze the relationship between psychological contract fulfillment and pharmacists’ attitudes and behaviors, and extend this discussion to the context of medication adherence management;

(3) With reference to relevant studies, we examine the mediating role of job burnout in the relationship between psychological contract and engagement in adherence management, and discuss potential underlying mechanisms;

(4) Finally, the Discussion concludes by outlining organizational-level practical implications and directions for future research, informed by the existing literature.

Together, these revisions constitute a comprehensive synthesis of prior research and clearly situate the present findings within the broader research context. The corresponding revisions have been fully incorporated into the Discussion section of the revised manuscript, and we believe this adequately addresses the reviewer’s comment.

Reviewer 2

Comments 1: This research is interesting, however, the presentation is a bit confusing. I suggest that the authors reformate it in a more systematic ways. So, this research contains two stages of research.The first one is the development and validation of the medication adherence management attitude and behaviour scale and the second one is using that scale to measure factors influencing it (psychological contract and job burn out).

Response 1: We thank you for your positive evaluation of this study and for your valuable suggestions. In response, we have systematically revised the Methods and Results sections by explicitly structuring the study into two phases. The first phase focused on the development and psychometric validation of a scale assessing hospital pharmacists’ attitudes and behaviors toward medication adherence management. The second phase applied the validated scale to examine factors influencing pharmacists’ participation in medication adherence management.

To enhance clarity and logical coherence, we have added the following statement to the Methods section: “This study was conducted in two phases. In the first phase, a scale measuring hospital pharmacists’ attitudes and behaviors toward medication adherence management was developed and psychometrically validated. In the second phase, the validated scale was applied to investigate factors influencing pharmacists’ participation in medication adherence management” (lines 87–89). Following this revision, both the Methods and Results sections are now presented according to these two phases, resulting in a clearer and more structured presentation of the study design. Importantly, these revisions did not involve any substantive changes to the study content or data analyses.

Comments 2: The authors should systematically write it as two stages in the methods and explained it clearly the methods used for each stages. The combination of this two stages of research has make presentation of the results become too much with all 12 tables, with some of them are very big tables. This was not effective for an article. This has also makes the explanation about the important variables measured and other important part of methods were missing. One example include there is no information about the instrument used for psychological contract and job burn out, and what it is actually about.

Response 2: We thank you for your valuable comments on the manuscript structure and presentation of the methods. In response to your suggestions, we have systematically revised the Methods and Results sections by explicitly dividing the study into two phases. The first phase involved the development and psychometric validation of a scale assessing pharmacists’ attitudes and behaviors toward medication adherence management. The second phase applied the validated scale to examine factors influencing pharmacists’ participation in medication adherence management. To improve clarity and logical flow, we added the following statement to the Methods section: “This study was conducted in two phases. In the first phase, a scale measuring hospital pharmacists’ attitudes and behaviors toward medication adherence management was developed and psychometrically validated. In the second phase, the validated scale was applied to investigate factors influencing pharmacists’ participation in medication adherence management” (lines 87–89). Following this revision, both the Methods and Results sections are now organized by study phase, resulting in a clearer and more structured presentation, while the study content and data analyses remain unchanged.

With regard to your suggestion to reduce the number of tables, we sincerely appreciate and fully understand the reviewer’s concern. However, each table included in the manuscript serves a distinct and necessary purpose, specifically presenting: (1) the results of the Delphi expert consultation, (2) the psychometric properties of the developed scale, (3) the analysis of the relationship between pharmacists’ psychological contract and medication adherence management, and (4) the mediation analysis examining the role of job burnout. Each table corresponds to a different stage of data processing or statistical analysis, and merging or removing tables may compromise the completeness and clarity of the reported results. Therefore, to ensure transparency and scientific rigor, we respectfully request that all tables be retained.

In addition, we have added a new subsection entitled “Survey Instrument” to the Methods section (lines 129–144), which provides a detailed description of the instruments used in the study, including the Hospital Pharmacists’ Psychological Contract Scale and the Job Burnout Scale. This subsection also clarifies the content and scoring methods of each instrument, addressing previously missing but essential methodological information.

Comments 3: The authors has mentioned about the theories on the psychological contract and job burnout with work outcomes. But has not been strong enough why part of the work outcomes is their attitude and behaviour in medication adherence management and not the other pharmacist job? The authors need to provide the context and the problem for this for example whether this job is a mandatory task for pharmacist in hospitals in the study locat

---

## [Decision Letter · Decision Letter 2]

10 Mar 2026

Dear Dr. Tang,

Thank you for submitting your manuscript to PLOS ONE. After careful consideration, we feel that it has merit but does not fully meet PLOS ONE’s publication criteria as it currently stands. Therefore, we invite you to submit a revised version of the manuscript that addresses the points raised during the review process.

We look forward to receiving your revised manuscript.

Kind regards,

Yaser Mohammed Al-Worafi

Academic Editor

PLOS One

Journal Requirements:

Reviewers' comments:

Reviewer's Responses to Questions

**Comments to the Author**

Reviewer #2: All comments have been addressed

Reviewer #4: All comments have been addressed

2. Is the manuscript technically sound, and do the data support the conclusions?

Reviewer #2: Yes

Reviewer #4: Yes

3. Has the statistical analysis been performed appropriately and rigorously?

Reviewer #2: Yes

Reviewer #4: Yes

4. Have the authors made all data underlying the findings in their manuscript fully available?

Reviewer #2: Yes

Reviewer #4: Yes

5. Is the manuscript presented in an intelligible fashion and written in standard English?

Reviewer #2: Yes

Reviewer #4: Yes

Reviewer #2: All previous points are addressed. Therefore, the research paper can be published. No further comments.

Reviewer #4: The manuscript addresses an important topic related to pharmacists’ engagement in medication adherence management; however, the introduction would benefit from a clearer articulation of the research gap and the practical relevance of studying psychological contracts specifically in relation to adherence management rather than other pharmacy services.

The study combines two distinct research stages (scale development and factor analysis) within a single manuscript, which makes the methodological presentation difficult to follow. The methods section should be reorganized to clearly separate the scale development phase from the analytical phase.

The manuscript includes a very large number of tables, several of which present extensive item-level information that may not be necessary in the main text. Consider moving detailed scale-development tables to supplementary material to improve readability.

More detailed information about the measurement instruments used for psychological contract and job burnout should be included earlier in the methods section, including their original sources, validity, and reliability in previous studies.

.

Reviewer #2: No

Reviewer #4: No

---

## [Author Response · Author response to Decision Letter 3]

7 Apr 2026

Dear Editors,

Thank you very much for your letter dated December 8, 2025, informing us about the revisions required for our manuscript titled “The impact of hospital pharmacists' psychological contracts on medication adherence management: the mediating role of job burnout” (Manuscript ID: PONE-D-25-16578). We sincerely appreciate the valuable guidance, comments, and suggestions provided by the reviewers.

We have carefully reviewed all the comments and revised the manuscript accordingly. All modifications have been highlighted in the manuscript for easy identification. Additionally, we have provided detailed point-by-point responses to the reviewers' comments below.

Reviewer #4

Comments 1: The manuscript addresses an important topic related to pharmacists’ engagement in medication adherence management; however, the introduction would benefit from a clearer articulation of the research gap and the practical relevance of studying psychological contracts specifically in relation to adherence management rather than other pharmacy services.

Response 1: Thank you for your valuable comment. Regarding the concern about the research gap and the rationale for focusing on medication adherence management rather than other pharmacy services, we have provided a structured explanation in the Introduction and further clarified and strengthened it in the revised manuscript.

First, we explicitly state in the Introduction that, within the current healthcare system in China, medication adherence management is generally not part of the formal job responsibilities of most pharmacists, but rather represents an additional, proactive professional service (see lines 49–63). Compared with traditional pharmacy services, such as prescription review and dispensing, adherence management is characterized as “non-mandatory but essential,” and its implementation relies heavily on pharmacists’ individual initiative.

Second, based on psychological contract theory, employees’ perceptions of the fulfillment of organizational obligations are known to influence their work motivation. Since adherence management is not strictly driven by formal job requirements but depends largely on voluntary effort, it is particularly important to explore the underlying psychological mechanisms. Accordingly, this study proposes that pharmacists’ psychological contract perceptions may influence their willingness to invest additional time and effort in adherence management, thereby establishing a theoretical link between psychological contract and this type of proactive service behavior.

Furthermore, drawing on social exchange theory and conservation of resources theory, we clarify the theoretical pathway through which psychological contract affects work behavior via job burnout (see lines 64–73). Prior research has also highlighted that healthcare professionals’ work engagement is closely associated with their performance in medication adherence management.

Therefore, this study focuses on medication adherence management as a specific context to examine how psychological contract influences pharmacists’ participation in “non-mandatory but critical” professional services. This approach helps address a gap in the literature, which has predominantly focused on general job performance while paying limited attention to such discretionary service behaviors. We believe that the revised Introduction now clearly articulates both the research gap and the practical relevance of the study.

Comments 2: The study combines two distinct research stages (scale development and factor analysis) within a single manuscript, which makes the methodological presentation difficult to follow. The methods section should be reorganized to clearly separate the scale development phase from the analytical phase.

Response 2: Thank you for your valuable comment. In response to your suggestion to clearly separate the two research stages in the Methods section, we have made systematic revisions to the manuscript structure in the previous round.

Specifically, we have explicitly stated in the Methods section that the study was conducted in two phases, with the following clarification added: “This study was conducted in two phases. In the first phase, a scale measuring hospital pharmacists’ attitudes and behaviors toward medication adherence management was developed and psychometrically validated. In the second phase, the validated scale was applied to investigate factors influencing pharmacists’ participation in medication adherence management.”

In addition, both the Methods and Results sections have been reorganized and presented according to these two phases, ensuring a clear structural distinction between the scale development process and the subsequent empirical analysis, and improving the overall coherence and clarity of the manuscript.

We believe that these revisions have substantially enhanced the logical flow and readability of the methodological presentation, and clearly reflect the design and implementation of the two research phases.

Comments 3:The manuscript includes a very large number of tables, several of which present extensive item-level information that may not be necessary in the main text. Consider moving detailed scale-development tables to supplementary material to improve readability.

Response 3: Thank you for your valuable suggestion. In response to your concern regarding the large number of tables and the inclusion of extensive item-level details, we have revised the manuscript accordingly.

Specifically, detailed item-level information and related tables from the scale development process have been moved to the Supplementary Material (see Supplementary Material file). Only the key result tables that are directly relevant to the main findings have been retained in the main text, in order to highlight the core results and improve the overall readability and structural clarity of the manuscript.

We believe that these revisions effectively address the reviewer’s concern regarding table presentation, while maintaining the completeness and transparency of the study.

Comments 4: More detailed information about the measurement instruments used for psychological contract and job burnout should be included earlier in the methods section, including their original sources, validity, and reliability in previous studies.

Response 4:Thank you for your valuable comment. In response to your suggestion, we have further supplemented the validity and reliability information of the measurement instruments in the Methods section.

Specifically, in the “Survey Instrument” subsection, we have added the Kaiser–Meyer–Olkin (KMO) values and Bartlett’s test of sphericity for the Hospital Pharmacists’ Psychological Contract Scale. The KMO values for the three subscales were 0.957, 0.930, and 0.917, respectively (all > 0.6), and Bartlett’s test was statistically significant (p < 0.001), indicating good construct validity. In addition, Cronbach’s α coefficients and split-half reliability coefficients were all greater than 0.6, demonstrating acceptable reliability.

For the Maslach Burnout Inventory, we have supplemented the internal consistency coefficients (Cronbach’s α) for its three dimensions, which were 0.89, 0.79, and 0.87, respectively, further supporting the reliability of the scale.

These additions provide more comprehensive information on the sources, structural characteristics, and psychometric properties of the instruments, and enhance the rigor and completeness of the Methods section.

We sincerely appreciate the constructive feedbacks from the editor and reviewers, which have greatly helped us improve our manuscript. We hope that the revised version meets the journal’s standards and look forward to your kind evaluation.

Best regards,

Fushan Tang

---

## [Editor Report · Decision Letter 3]

8 Apr 2026

The Impact of Hospital Pharmacists' Psychological Contracts on Medication Adherence Management: The Mediating Role of Job Burnout

PONE-D-25-16578R3

Dear Dr. Tang,

We’re pleased to inform you that your manuscript has been judged scientifically suitable for publication and will be formally accepted for publication once it meets all outstanding technical requirements.

Kind regards,

Yaser Mohammed Al-Worafi

Academic Editor

PLOS One
---

## [Editor Report · Acceptance letter]

PONE-D-25-16578R3

PLOS One

Dear Dr. Tang,

I'm pleased to inform you that your manuscript has been deemed suitable for publication in PLOS One. Congratulations! Your manuscript is now being handed over to our production team.

Kind regards,

on behalf of

Professor Yaser Mohammed Al-Worafi

Academic Editor

PLOS One